# Predictors of Students’ Mental Health during the COVID-19 Pandemic: The Impact of Coping Strategies, Sense of Coherence, and Social Support

**DOI:** 10.3390/ijerph192416423

**Published:** 2022-12-07

**Authors:** Charlotte Torinomi, Katajun Lindenberg, Andreas Möltner, Sabine C. Herpertz, Rainer M. Holm-Hadulla

**Affiliations:** 1Department of Child and Adolescent Psychotherapy, Institute of Psychology, Goethe-University Frankfurt, 60486 Frankfurt, Germany; 2Dean’s Office, Medical Faculty, Heidelberg University, 69120 Heidelberg, Germany; 3Department of General Psychiatry, Centre for Psychosocial Medicine, Heidelberg University Hospital, 69115 Heidelberg, Germany

**Keywords:** COVID-19 pandemic, depression, mental health, coping, sense of coherence, social support, resilience, university students

## Abstract

Young people and women seem to suffer more from social restrictions due to the COVID-19 pandemic than do others. Findings from pre-pandemic surveys identified students as a specific risk group for developing anxiety and depressive symptoms. Recent studies have indicated that students especially denoted a decrease in mental health during the pandemic. In a sample of *n* = 1938 university students (67.6% female), we investigated protective factors that are associated with mental health (defined as the absence of any mental disorder) and more specifically, the absence of major depression during the pandemic despite social restrictions. Investigated protective factors were social support, sense of coherence and situational coping strategies. The results of the multiple logistic regression analyses revealed that male gender, high sense of coherence and specific coping strategies could be identified to be associated with mental health in general and the absence of major depression. Protective coping strategies that were related to mental health in general were lower substance use, lower behavioral disengagement, higher positive reframing and lower self-blame. Protective coping strategies that were associated with the absence of major depression specifically were higher use of instrumental support, lower substance use, lower behavioral disengagement, higher positive reframing, higher emotional support, lower self-blame and lower humor. Social support was related to the absence of major depression, but not to mental health in general. Higher age in university students was associated with better mental health, but not with the absence of major depression specifically. These findings indicate that sense of coherence and situational coping strategies can buffer the adverse effect of social restrictions on mental health and thus, can serve as important resilience factors. Moreover, they highlight the political relevance of promoting specific coping strategies to foster mental health in students encompassing adverse events and social restrictions.

## 1. Introduction

Social restrictions were initiated in many countries all over the world with the beginning of the COVID-19 pandemic. There is growing evidence showing that this was associated with significantly higher numbers of mental health problems. In a representative German survey [1], which investigated mental health during the COVID-19 pandemic compared to a representative pre-pandemic assessment in 2018, symptoms of depression and anxiety significantly increased. Women and young people, especially, seemed to be affected more than other groups of age or gender [1]. In a random effect meta-analysis of U.S. and Chinese data bases, Deng et al. (2021) found elevated prevalence of anxiety and depressive symptoms compared to the pre-pandemic status [2]. An increase in mental health problems was found in a longitudinal study in the U.K. in early stages of the pandemic compared to previous years [3]. Daly et al. (2021) analyzed longitudinal data from two nation-wide U.S. surveys, the National Health and Nutrition Examination Survey (NHANES) and the Understanding America Study (UAS), concerning effects of the COVID-19 pandemic on the prevalence of depressive symptoms. The sample characteristics were comparable. The analysis of NHANES data indicated that the prevalence of depressive symptoms was relatively stable across the U.S. from 2007 to 2018. Therefore, a comparison between the last wave of NHANES from 2017 to 2018 and from March as well as April 2020 was conducted. A significant increase in depressive symptoms, measured with the Patient Health Questionnaire-2 (PHQ-2), was identified compared to 2018, with 8.7% to 10.6% in March 2020 and 14.4% in April 2020. The 18–34 aged subgroup denoted the highest increase in depressive symptoms compared to the other subgroups [4]. Students, in particular, seemed to develop feelings of uncertainty and loneliness including high occurrence of depressive symptoms, anxiety and stress during the pandemic [2]. In line with these findings, an Australian study [5] reported a deterioration in mental well-being since COVID-19 onset in 68% of college students.

### 1.1. Students’ Mental Health during the Pandemic

Before the pandemic in 2014/15, Weber et al. determined that 53.6% of the students at the University of Cologne (Germany) showed mental health problems and 20.6% showed major depression symptoms measured with the Patient Health Questionnaire (PHQ), which is widely used all over the world to screen for psychiatric symptoms and diagnoses [6]. Using the same method during the pandemic in spring 2021 [7], much higher rates of mental health problems were reported at Heidelberg University (Germany). In this survey of 2349 students, a total of 75.8% showed mental health problems screened with the German version of the Patient Health Questionnaire (PHQ-D). Clinically relevant symptoms of major depression were especially high at 41.6%.

During the lockdown, opportunities for social interaction at the university seemed to have had a protective effect against psychological stress. Students of Law, Economics and Social Sciences, Humanities and Theology suffered more from pandemic circumstances than did students of Medicine or MCNT (Mathematics, Computer Science, Natural Science, Technical Studies) [7]. These students still participated in face-to-face events of classroom teaching, patient teaching or laboratory teaching. An analysis of 1089 free reports underlined the dominance of depressive symptoms in students (such as depressive moods, lack of drive and motivation, hopelessness, and most frequently reported, loneliness). Students frequently reported that the above-mentioned symptoms were primarily driven by social contact restrictions. They made suggestions to improve their situation including providing opportunities for face-to-face studies and social interactions at the university [7].

An increase in mental health problems during the pandemic has also been reported from China [8], where 746,217 students completed parts of the PHQ questionnaire in February 2020. Of these, 45% reported mental health problems, and 21.1% of them showed depressive symptoms [8]. In the SERU COVID-19 survey, about 45,000 undergraduate, graduate, and professional students at nine U.S. universities were screened with the PHQ-2 for depressive symptoms, and with the GAD-2 for anxiety symptoms between May and July 2020. A total of 35% of undergraduate and 32% of graduate and professional students showed severe depressive symptoms. Compared to data from a study with graduate and professional students in 2019, the number of students in this group screening positively for depressive symptoms was twice as high (2019: 15%; 2020: 32%) [9].

Young adults, especially university students, seem to be notably vulnerable to developing mental health problems such as depression or anxiety disorders due to various transitions, uncertainties and high demands on self-regulation [10,11]. Consequences of the COVID-19 pandemic might have multiplied these transitional challenges for university students. The COVID-19 pandemic can be considered to be an adverse event that requires specific individual adaptation. The “dynamic process encompassing positive adaption” [12] has been referred to as “resilience”. Resilience is defined as the counterpart of vulnerability and evolves during stages of development throughout the life span [13]. It can be described as hardiness against the impact of adverse life circumstances or risk factors, and furthermore, the process of developing successful coping strategies [14]. For the development of resilience in children being confronted with adverse circumstances, the following aspects were found to be relevant: emotion regulation, family related factors, peers and network factors, and cultural-societal factors [13]. Resilience and protective factors have frequently been investigated in adult individuals after experiencing adverse events. As an example, after the terrorist attacks on 11 September 2001, higher well-being and lower distress were predicted by resilience factors such as higher education, less emotional suppression, fewer negative worldview changes, social support and coping strategies [15]. Based on those findings, this study focuses on individual and social protective resilience factors in young individuals.

### 1.2. Social Support

A number of protective factors on the individual and social levels have been identified as buffering the effects of transitional stress on mental health in university students. Social support has been found to have a significant protective effect against internalizing disorders in university students. In addition, less loneliness, which is associated with higher social support, seems to protect from anxiety, depression or substance abuse [8,16,17]. During the pandemic, feelings of loneliness increased, as well as mental health problems in general, especially in young adults [18]. Ma et al. showed in their study of college students during the pandemic that lower perceived social support augments the probability of having depressive symptoms by 4.84 to 5.98 times. [8] Although social support might serve as a protective factor over the whole life span, its effect might be of particular importance in transition stages [19]. Social support can be subdivided into instrumental, informational, and emotional support. Instrumental support includes concrete actions such as helping with housekeeping or shopping. Informational support means to provide important information (e.g., announcements) about opportunities to act, while emotional support can include, e.g., listening and provide encouragement [20]. In a systematic review, Gariépy et al. found emotional support to be the most consistent part of social support that prevents depressive symptoms in younger adulthood (75% of the included studies), followed by instrumental support (67% of the included studies) [18]. These findings underscore that social support can be seen as a social protective factor for mental health.

### 1.3. Sense of Coherence

The sense of coherence (SOC) based on Antonovsky’s model of salutogenesis describes a relatively stable personality trait. The salutogenetic model focuses on how to actively conserve a healthy state. The amount of SOC describes how individuals cope with stress and life challenges. It consists of three dimensions: comprehensibility, which is described as the ability to recognize stressors as structured, predictable and consistent; manageability, which is the belief of having sufficient resources to face the present stressors; meaningfulness, which is defined as the belief that the stressor is worth investing energy in [21,22,23,24]. A distinct but related construct is the self-efficacy of Bandura (1977) [25], which was developed further as a stable general trait by Schwarzer and Jerusalem (1995). High self-efficacy refers to being convinced of being able to overcome adverse circumstances through specific behaviors [26]. The manageability dimension of SOC overlaps with self-efficacy. In a longitudinal study with three times of measurement, including the ages of 15, 16, and 19, SOC was the strongest predictor at the age of 16. At the age of 19, it was the only predictor of psychological symptoms except for previous psychological symptoms. General self-efficacy did not predict mental health [27]. Therefore, SOC was chosen as a potentially relevant factor instead of general self-efficacy. SOC has also been found to play a decisive role in maintaining students’ mental health and in protecting from depressive symptoms in pre-pandemic studies [23,24,28]. For example, Ying et al. found SOC to have a mediating role between college demands and developing depressive symptoms in Chinese–American college students [23]. In addition, SOC dimensions and their impact on mental well-being were assessed during the COVID-19 pandemic, e.g., [29]. Results from previous studies were confirmed. Consequently, a high sense of coherence seems to be a protective trait factor for mental health.

### 1.4. Coping Strategies

In this study, the coping model of Carver, Scheier, and Weintraub was assessed. In contrast to various dichotomous coping models, Carver et al. (1989) postulated a multidimensional approach [30]. It combines the self-regulation model of Carver and Scheier as well as the transactional model of stress and coping of Lazarus and Folkman (1984) [31]. They define coping as behavioral and cognitive efforts to manage stressful events or circumstances. Carver et al. (1989) proposed to not categorize most of the coping strategies as strictly helpful or not [30]. Coping strategies can be seen as specific behaviors to overcome an adverse event, while emotion regulation strategies are not specifically related to an adverse event. They are helpful in handling positive and negative emotions. Therefore, this study focused on coping strategies instead of emotion regulation [32].

Mental well-being seems to be influenced by individuals’ coping strategies in specific situations [33,34]. In a study by Chao and colleagues, problem-focused coping mediated the moderating effect of social support on psychological well-being and perceived stress in students. Avoidant coping aggravated the relationship between stress and well-being [33]. In a pre-pandemic study, Kurtovic et al. found higher situational problem-focused coping and active coping, e.g., finding solutions for a problem, to predict a lower level of psychopathological symptoms including depressive and anxiety symptoms. At the same time, emotion-focused coping was predictive of higher symptom levels [34]. Coping strategies have also been investigated as protective or risk factors during the COVID-19 pandemic. In the survey of Prowse et al., it was shown that students tended to cope with the adverse events during the pandemic with substance use or social media use, which were related to a negative impact on mental health [9]. In a Polish study with students from four universities, results indicated that the younger the students were, the less coping strategies they used. The most-used coping strategies during the lockdown in the Polish study were acceptance, planning, and seeking emotional support, while in other countries other coping strategies are more frequently used to cope with stress [35,36,37]. The authors point out the risks to mental health that result when maladaptive coping strategies are used in the long-term. Generally, coping strategies can be seen as a situational protective or risk factor depending on the individual coping strategy and if it is used successfully.

Although social support, sense of coherence and coping strategies seem to be important factors in fostering mental health in students, little is known about the protective nature of these social factors, individual traits and situational factors during a persisting adverse event and corresponding social restrictions. 

### 1.5. Hypotheses

We wanted to investigate students’ mental health during the COVID-19 pandemic and identify factors that predict mental well-being despite the occurrence of a persisting adverse event and social restrictions. We raised the hypotheses that age (H1a), gender (H2a), social support (social protective factor) (H3a), sense of coherence (individual trait) (H4a), or individual coping strategies in specific situations (such as active coping (H5a_1)), planning (H5a_2), positive reframing (H5a_3), acceptance (H5a_4), humor (H5a_5), religion (H5a_6), use of emotional support (H5a_7), use of instrumental support (H5a_8), self-distraction (H5a_9), denial (H5a_10), venting (H5a_11), substance use (H5a_12), behavioral disengagement (H5a_13), and self-blame (H5a_14) are strongly associated with mental well-being in students during the pandemic. We also wanted to identify protective factors that were associated with the absence of major depression. We examined whether age (H1b), gender (H2b), social support (H3b), sense of coherence (H4b), or single situational coping strategies (active coping (H5b_1)), planning (H5b_2), positive reframing (H5b_3), acceptance (H5b_4), humor(H5b_5), religion (H5b_6), use of emotional support (H5b_7), use of instrumental support (H5b_8), self-distraction (H5b_9), denial (H5b_10)l, venting (H5b_11), substance use (H5b_12), behavioral disengagement (H5b_13), and self-blame (H5b_14) were associated with a major depression syndrome.

## 2. Materials and Methods

### 2.1. Procedure

This study is based on the German study “Well-being, Mental Health and Coping Strategies of students in times of COVID-19 pandemic” (original title: “Wohlbefinden, mentale Gesundheit und Copingstrategien von Studierenden in Zeiten der COVID-19 Pandemie”). Data collection was conducted between 26 May 2021 and 11 June 2021 using the platform “Limesurvey”. A total of *n* = 27,162 university students at Heidelberg University (Germany) holding university email addresses were contacted. At that time, students had been faced with restrictions at the university (e.g., mostly digital lessons; libraries, refectories, cafeterias and sports facilities were widely closed) for about 1.5 years. The response rate was 8.83% (*n* = 2308) including surveys with missing responses. On the one hand, the recruitment letter might have had an impact on the low response rate because it might have motivated students who were interested in finding out how to cope with the pandemic circumstances and who felt a decline in well-being at that time. On the other hand, feeling overstrained might have inhibited motivation to participate in the survey.

After listwise deletion, a total of 7.26% (*n* = 1973) complete datasets remained. It was observable that the number of participants decreased with the chronological order of the questionnaires; the later the questionnaire was presented, the fewer participants completed it (PHQ-D: *n* = 2139; Brief COPE: *n* = 2028; SOC-13: *n* = 1981, ESSI-D: *n* = 1973). This might indicate that the protocol was too long, especially for strained students.

For the regression analyses, the gender category “divers” or “not specified” was excluded due to the small number of cases (*n* = 1938). The study design was approved by the data security officer of Heidelberg University and the ethics review committee of University Hospital Heidelberg for the original data collection. Primary and secondary analyses were covered by an ethical vote. The anonymity of the participants was guaranteed. Informed consent was obtained, and participation was voluntary, including the option to drop out without giving reasons at any time.

### 2.2. Measures

The following investigative tools were applied as part of the primary analysis: the Patient Health Questionnaire (PHQ-D), the German versions of the WHO-5 Well-Being-Index, the German non-validated adaptation of the Fear of COVID-19 Scale (FCV-19S), and an open question format part concerning feelings about the COVID-19 pandemic and how the students dealt with the situation. The WHO-5-Well-Being-Index is able to identify the absence or presence of major depressive syndromes, especially mild and moderate types, but is not able to identify severe depressive syndromes or general psychopathology [38]. For this reason, the results of the Patient Health Questionnaire (PHQ-D) were chosen as the criteria variable for the secondary analysis.

In addition, the German SPF (Saarbrücker Persönlichkeitsfragebogen based on the Interpersonal Reactivity Index (IRI)), which measures different independent components of empathy [39], and the German version of the General Self-Efficacy Scale (SGSE), which conduces how much a person is convinced they can cope with an adverse event and influence it [27], were assessed but were not part of the secondary analysis. The IRI was excluded from the secondary analysis because of the assumption that empathy might be more relevant in helping jobs, e.g., in the medical sector. Cross-sectional studies indicate a correlation between empathy and mental health as well as empathy and professional competencies, but there is a lack of longitudinal studies [40]. In this sample, empathy might be relevant for a small number of students. Therefore, the IRI was excluded from the secondary analysis.

Mental health was measured with the German adaptation of the Patient Health Questionnaire (PHQ-D) [41]. The PHQ-D is a screening instrument to assess somatoform syndrome (8 items), depressive syndromes (9 items, PHQ-9), panic (15 items), other anxiety syndromes (6 items), bulimic symptoms and binge eating symptoms (8 items), alcohol use (6 items), and general stress including stress factors in non-psychiatric patients that might be trigger and sustentative factors for developing mental health problems (24 items). The PHQ is internationally the most frequently used instrument to screen depressive, anxiety and somatoform syndromes, and was developed for general medical practitioners who are not psychiatrists. In this study, we focused on data analyses regarding general mental health (absence of meeting any PHQ-syndrome) and major depression specifically. The criteria for meeting clinically relevant syndromes are specified in the PHQ-D manual and calculated as follows: somatoform syndrome—minimum of three items are answered with “strongly impaired” and absence of adequate organic cause; major depressive syndrome—five or more items are answered with “more than half the days” or “nearly every day” including the items “little interest or pleasure in doing things” and “feeling down, depressed, or hopeless” (the item “thoughts that you would be better off dead or of hurting yourself in some way” is included if it is answered with “several days”); other depressive syndromes—two to four items are answered with “more than half the days” or “nearly every day” including the items “little interest or pleasure in doing things” and “feeling down, depressed, or hopeless” (the item “thoughts that you would be better off dead or of hurting yourself in some way” is included if it is answered with “several days”); panic—first, the answer “yes” is given for the items “Have you had an anxiety attack during the last 4 weeks (instant feeling of fear or panic)?”, “Had that ever happened before?”, “Do any of these attacks appear unexpectedly in situations you do not expect to react strained or alarmed?”, “Do you feel strongly impaired by those attacks and/or are you afraid of having another attack?”, and second, four or more physiological symptoms occurred during an attack of anxiety and/or the fear of dying; other anxiety syndromes—the item “How often have you felt nervous, anxious, or on edge during the last four weeks?” and three or more further items are answered with “more than half the days”; suspected bulimia nervosa and suspected binge eating disorder—the answer “yes” is given for the items “Do you often have the feeling that you are not able to control how much you eat or what you eat?”, “Do you often eat so much within 2 h that others would see as a considerable quantity?”, and for suspected bulimia nervosa, if the item “if you used minimum one of weight regulation strategies do you use it on average twice a week?” is answered with yes, for binge eating disorder, if this item is answered with “no”; alcohol use—if one item is answered with “yes”; general stress and stress factors—a sum score is calculated (0–20; 0 = “Not impaired”, 1 = “Rarely impaired”, 2 = “Severely impaired”). In addition, sum scores can be calculated for depressive syndromes using the depressive syndrome scale (PHQ-9) including classification of the level of severity (0 = “Not at all”, 1 = “Several days”, 2 = “More than half the days, 3 = “Nearly every day”; 5–9 mild type, ≥10 major depression, 10–14 moderate type, 15–19 severe type, 20–27 most severe type). It is possible to calculate sum scores for somatic symptoms (consisting of the somatoform syndrome scale plus two items of the depressive syndrome scale = PHQ-15). For further analyses, criteria were dummy coded (1 = meeting criteria for the specific syndrome, 0 = absence of specific syndrome). Mental health was defined as the absence of any PHQ-syndrome. The absence of major depression was defined as not meeting the full criteria for the PHQ-major depressive syndrome (scores 10 or higher) [41].

Social support was assessed with the German version of the ENRICHD Social Support Inventory (ESSI-D). The ESSI-D consists of seven items (six items with a five-stage response format (1 = “None of the time” to 5 = “All the time”), as well as one item asking about living with a partner or being married. The questionnaire provides one score and asks about different kinds of social support as structural, instrumental, and emotional support (e.g., “Is there someone available to you who you can count on to listen when you need to talk?“, “Can you count on anyone to provide you with emotional support, such as talking over or helping you make difficult decisions?“, “Do you have as much contact as you would like with someone you feel close to, someone you can trust and confide in?”) [42]. The German adaptation shows good psychometric properties and is an efficient instrument to measure perceived social support [43]. It was originally created for patients with chronic somatic illnesses.

The Sense of Coherence Scale is based on Antonovsky’s concept of salutogenesis [21]. He postulated a continuum between illness and health. The extent of sense of coherence indicates the position of an individual on the continuum. A higher sense of coherence predicts being and staying healthy. The questionnaire consists of the sub scales comprehensibility (five items), manageability (four items), and meaningfulness (four items). Items are processed with a double pole seven-stage response format. A sum score is calculated. In this study the German short version (SOC-13) was used [22].

Coping with stress was assessed with the Brief COPE-Inventory (Coping Orientation to Problems experienced) focusing on strategies for dealing with stress. It is based on the 60-item COPE-Inventory. Theoretically, it is founded on the coping theory of Lazarus and Folkman and partly on the behavioral self-regulation theory by Carver and Scheier (1990) [26,44,45]. Due to time constraints, the brief version of the COPE was used. It is a multidimensional instrument with 14 two-item factors, including active coping, planning, positive reframing, acceptance, humor, religion, use of emotional support, use of instrumental support, self-distraction, denial, venting, substance use, behavioral disengagement, and self-blame. The questionnaire consists of 28 items with a four-stage response format (1 = “I haven’t been doing this at all” to 4 = “I’ve been doing this a lot”). As recommended by Hanfstingl et al. (2021), who tested the factor structure, the original 14 factor structure was kept instead of a higher structure order [46]. Because of the exceptional circumstances of the pandemic, retrospective situational coping was assessed instead of concurrent situational coping or dispositional coping [47]. These can be differentiated by formulating items in different tenses. In our study, items were expressed in past perfect instead of present tense [45].

Reliability indexes of all deployed questionnaires are provided in Table 1.

### 2.3. Statistical Analyses

This study was a secondary analysis of existing data. Data analysis was run through R version 4.1.0. Demographic variables were coded as follows: age (1 < 21 years, 2 = 21–23 years, 3 = 24–25 years, 4 = 26–27 years, 5 >27 years); gender (2 = female, 3 = male); field of study (1 = Medicine, 2 = Law, 3 = Psychology, 4 = Economics & Empirical Social Sciences, 5 = MCNT–Mathematics, Computer Science, Natural Science, Technical Studies, 6 = Humanities & Theology, 99 Others). Descriptive statistics (mean values, standard deviations, and reliability coefficients) of all included variables are presented in Table 1. We conducted analyses for (a) identifying variables which are associated with mental health in general (coding: 1 = no PHQ-syndrome, 0 = any PHQ-syndrome), and (b) with the absence of major depression (coding: 1 = no PHQ-major depression, 0 = PHQ-major depression). For both outcomes, we conducted binary logistic regression analyses in the first step and a multiple logistic regression analysis in the second step. Predictors were age, gender, social support, sense of coherence and coping strategies (active coping, planning, positive reframing, acceptance, humor, religion, use of emotional support, use of instrumental support, self-distraction, denial, venting, substance use, behavioral disengagement, and self-blame). To monitor multicollinearity in the model, variance inflation factors (VIF) were calculated for the independent variables (range between 1.036 and 2.646). Hence, multicollinearity did not occur.

## 3. Results

### 3.1. Descriptive Statistics

The sample with complete datasets without any unanswered items (*n*= 1973) consisted of 627 male (31.27%) and 1311 female (66.45%) students. The gender category “divers” or “not specified” was reported by 35 students (1.77%). To calculate scale means and standard deviations, listwise deletion was performed (see Table 1). More than two-thirds of the sample were 23 years old or younger (<21 years, 27.01%; 21–23 years, 39.64%; 24–25 years, 16.22%; 26–27 years, 6.89%; >27 years, 10.24%). Fields of study were distributed as follows: MCNT, 30.97%; Humanities and Theology, 24.13%; Medicine, 15.86%; Economics and Social Sciences, 10.59%; Law, 9.28%; Psychology, 1.57%; others, 7.60%.

### 3.2. Bivariate Logistic Regression Analyses

In the bivariate regressions analyses, we found higher age, male gender, more social support, higher sense of coherence and various coping strategies predicted (a) mental health in general (i.e., the absence of any PHQ-syndrome, see Table 2) and (b) the absence of major depression (i.e., the absence of PHQ-depression syndrome, see Table 3). Protective coping strategies were higher active coping, higher positive reframing, higher acceptance, higher emotional support, higher self-distraction, lower denial, lower substance use, lower behavioral disengagement and lower self-blame. In addition, the use of instrumental support was associated with the absence of major depression, but not with higher mental health in general.

### 3.3. Multiple Logistic Regression Analyses

The results of the multiple logistic regression analyses confirmed that male gender (H1a), more social support (H3a and b), higher sense of coherence (H4a and b) and situational coping strategies were independently associated with (a) mental health in general (see Table 2) and (b) the absence of major depression (see Table 3). Coping strategies that were related to mental health in general were higher positive reframing (H5a_3), lower substance use (H5a_12), lower behavioral disengagement (H5a_13), and lower self-blame (H5a_14) (Table 2). Coping strategies that were associated with the absence of major depression, specifically, were higher positive reframing (H5b_3), higher emotional support (H5b_7), higher use of instrumental support (H5b_8), lower humor (H5a_ 5), lower substance use (H5b_12), lower behavioral disengagement (H5b_13), and lower self-blame (H5a_14) (Table 3). Higher age among university students was significantly associated with higher mental health, but not with major depression.

## 4. Discussion

Our results show widely identical resilience factors related to the absence of mental health problems and the absence of a major depression syndrome during the pandemic. Stable demographic and trait factors that are associated with mental health in general, or the absence of major depression syndromes, in particular, are male gender and high sense of coherence. Coping strategies such as higher positive reframing, lower self-blame, lower substance use, and lower behavioral disengagement appear to be situational protective factors. The fact that these factors seem to be disorder-unspecific and might protect against both general psychopathology and against depression specifically, must be interpreted in light of the fact that depression was also found to be the most common disorder group in this sample.

Lower age among university students appeared to be a risk factor for a decline in mental health in general, but not for developing a major depression syndrome. In contrast, higher use of emotional support and higher social support seem to be situational social protective factors for not developing major depression syndromes specifically, but not for general psychopathology. These results correspond partly with resilience theory; positive reframing is a coping strategy that is associated with adaptive emotion regulation. At the same time, self-blame is associated with maladaptive emotion regulation [13,14,32]. It is likely that high SOC could be the predictor for successful coping, and it seems to play a significant role for mental health in general. Social and emotional support were expected to be relevant for mental health, but they were not in this study.

The time of entry into university during the pandemic can be seen as a critical point for the development of a mental disorder in general, as it is a critical time of transition into young adulthood, when high demands are placed on self-regulation and high academic achievement. Kugbey et al. (2015) found social support of friends and significant others to be relevant for the level of depression in college students, while family support seemed to play an important role for perceived stress but not for the level of depression. We did not examine the sources of social support. There might be different effects on mental health in general or other mental disorders [48]. Especially, contact with peers and friends was highly restricted in Germany because people were only allowed to meet with more people if they lived in the same household, which was mostly not the case with friends. In contrast, depressive syndromes seemed to occur equally often throughout the duration of the study. Due to the fact that social support is associated with developing a depressive syndrome, social restrictions during the COVID-19 pandemic that inhibited meeting people in person, including meeting new people, might enhance the probability of developing depressive syndromes. The coping strategy “use of instrumental support” (“I’ve been trying to get advice or help from other people about what to do” and “I’ve been getting help and advice from other people”), which might be a protective factor, highlights the relevance of this sub-area of social support, which is also asked in the ESSI-D (“Is there someone available to give you good advice about a problem?”) [41].

Planning, religion, and venting did not play a significant role in mental health or the absence of a major depressive syndrome during the social restrictions of the COVID-19 pandemic in our study. These findings may result from weak scale properties on the one hand (cf. Table 1). On the other hand, we did not inquire about the religiousness of the students. Consequently, we are not able to estimate if religion plays a significant role in the life of the student sample. According to the results of Carver et al. (1989), when they developed the original COPE-Questionnaire, different findings about the frequency of using single coping strategies were reported. If situational coping was examined, higher engagement in planning was reported if the person estimated the event or situation as modifiable instead of something that had to be accepted [29]. We can argue that the pandemic circumstances at the university were not changeable and conditional by policy. This might have led to a reduced usage of strategy planning. If a strategy is less used it is not able to prevent mental strain. Venting as a situational coping strategy seems to be often used if the event matters to the person, as Carver et al. stated [29]. They argue that venting might be a helpful strategy on one hand, e.g., to move on after the loss of a close person, but on the other hand, it can hinder adjustment to distress if distress is emphasized and maintained, e.g., in the sense of rumination [29]. It could be seen as a maintaining factor. For example, Marr et al. (2022) found venting to be a mediating variable concerning chronic general anxiety disorder and 18 years later, major depressive disorder and vice versa [49]. But it should be taken into account that relevant dispositional coping strategies are not congruent with relevant situational coping strategies that also might have an impact on mental health or the absence of major depression.

Moderating effects might be responsible for the significant finding in the multiple logistic regression analyses but not in the bivariate analyses and change of algebraic sign in the multiple logistic regression analysis compared to the bivariate analysis. This might have been the case for the situational coping strategy, humor, and make it difficult to interpret. Lower humor was associated with the absence of major depression exclusively in the multiple logistic regression analysis, so it is not interpretable. In a Swiss study, results showed that the effectiveness of coping with stressful events and resulting negative emotions depends on the type of humor. If humor is positive, it has a positive impact on down- regulating negative emotions and fostering positive emotions. If humor is mean and negative, the converse effect was found [50]. In the Brief COPE, the items do not specify the type of jokes or making fun about the situation. It is possible that the students used a more detrimental style of humor.

Significant findings in the bivariate analyses but not in the multiple logistic regression analyses (e.g., active coping, acceptance) might be explained by stronger third variables in the model that correlate with the dependent variable. There is a correlation of the weaker variables with the outcome, but there seem to be other variables with a better fit so that the weaker variables might consequently be less relevant than others.

There are several studies that focus on mental health of students during the pandemic, but only a few studies have investigated students’ coping strategies, e.g., [11,36]. None of them analyzed which coping strategies are predictors for mental health or the absence of major depression in combination with perceived social support and the extent of sense of coherence. The findings of the survey of Salman et al. (2020) and El-Monshed et al. (2021), who used comparable instruments in a Pakistani sample and in an Egyptian sample of university students, support our results but differ in some critical respects [36,37]. Salman et al. found a remarkable number of students reporting anxiety with about 34% and depression with about 45% measured with the PHQ, which is comparable to the findings of our first study [7]. Furthermore, younger age was a risk factor for developing major depression, which corresponds to our results, and which also points to female gender as a risk factor for suffering more from social restrictions compared to male students. The relevance of single coping strategies differed in their study compared with our results. In contrast to our survey, Salman et al. found religious coping to be deployed by the majority of the sample, followed by acceptance, self-distraction, and active coping. These students tended to adopt more useful coping strategies [36]. In the Egyptian sample, 11.3% reported severe depressive symptoms and 15.4% reported extremely severe depressive symptoms. Loneliness was often reported as severe. High loneliness was a significant predictor for depression. Helpful functional coping strategies were negatively correlated with depression, while dysfunctional coping strategies were positively correlated with depression. Students reported the most frequent use of planning and active coping. In contrast, venting, denial, and substance use were utilized less [32]. Differences in religiosity and cultural background could also play a significant role in explaining the differences between our results and those of Salman et al. (2020) and El-Monshed et al. (2021) [36,37]. This study has some limitations. First, it was an online survey. Consequently, selective participation occurred. In addition, diverse biases are associated with self-report measurements as, e.g., systematic response [51] and underreporting of depressive symptoms [52]. On the other hand, students may have taken part if they felt affected by the restrictions. The low response rates of 8.83% and 7.26% complete datasets might indicate that, which raises the question of whether the results are conferrable to all students.

Reliability scores of some of the Brief COPE scales (planning, venting, acceptance, self-distraction, and denial) are partly less robust due to two-item scales. In addition, Huckins et al. found in a longitudinal and ecological momentary assessment (EMA) study, students’ mental strain varies through academic terms and is associated with academic demands and exam periods [53]. They monitored sleep, sedentary time, depressive and anxiety symptoms measured with the PHQ-4 in a student App. Mental health and behavioral changes were associated with the COVID-19 outbreak and increasing social restrictions in undergraduate students. Depressive and anxiety symptoms increased at that point and remained high, especially compared to previous academic terms. This study was not able to map seasonal trends due to its cross-sectional design.

Although data were not collected in an exam period, higher demands on self-regulated learning and not being adequately prepared for digital lessons and learning might have influenced mental health [54]. We did not monitor PTSD or adjustment disorder symptoms, which might occur due to loss of close family members or seeing them suffer from severe disease, or to expose oneself to pictures and videos of people dying or suffering heavily from COVID-19. In Italy, an increase in PTSD prevalence was found compared to pre-pandemic times [55].

Some further research is needed. As we know from the results of our first study [7], that the decrease in mental health, the extent of psychopathological symptoms, and worries about social restrictions varies across academic disciplines. Students of Medicine or MCNT reported fewer mental health complaints. For this reason, further analyses could reveal differences between students of different fields of study, including how they cope, the extent of perceived social support and how they claim it, and if they are well grounded in the dimensions of sense of coherence. In addition, it could be investigated if and how dimensions such as comprehensibility, manageability, and meaningfulness have been of importance for mental health during the COVID-19-pandemic. The German Study Co. II indicates that low socio-economic background and pre-pandemic psychiatric diagnoses may be risk factors for mental health complaints during the COVID-19 pandemic [54]. These aspects were not investigated in our study.

Focusing on fears concerning family members or friends diagnosed with COVID-19 might have had an impact on mental health as well. A one-year follow up study might clarify the trend of severity of psychopathological symptoms in students after having returned back to face-to-face teaching and social life on campus. Long-term effects on mental health should be investigated.

## 5. Conclusions

The findings of the present survey indicate that students heavily suffer from social restrictions and that pandemic policy severely affects private and professional life. A high prevalence of depressive and anxiety syndromes is apparent. It indicates the necessity of counselling services and preventive training to cope with the lack of social interactions. Pandemic policy should consider the mental health complaints of students and consequently reduce social isolation. Preventive measures should provide students with helpful coping strategies to improve academic and self-regulation as well as social support.

## Figures and Tables

**Table 1 ijerph-19-16423-t001:** Descriptive statistics and reliability indices (Cronbach’s Alpha) of mental health, social support, sense of coherence and coping strategies (*n* = 1973).

Variable	Mean (SD)	Cronbach’s Alpha
Major depression (PHQ-9)	11.68 (6.09)	0.87
Social support (ESSI total score)	19.96 (4.76)	0.92
Sense of coherence (SOC total score)	53.65 (12.48)	0.82
Coping (Brief COPE)		
Active Coping	2.46 (0.71)	0.76
Planning	2.79 (0.71)	0.45
Positive Reframing	2.3 (0.8)	0.74
Acceptance	2.43 (0.76)	0.66
Humor	2.11 (0.87)	0.78
Religion	1.56 (0.84)	0.84
Use of Emotional Support	2.58 (0.85)	0.82
Use of Instrumental Support	2.38 (0.9)	0.89
Self-Distraction	2.57 (0.72)	0.50
Denial	1.47 (0.66)	0.62
Venting	2.08 (0.77)	0.63
Substance Use	1.31 (0.62)	0.90
Behavioral Disengagement	1.57 (0.62)	0.51
Self-Blame	2.46 (0.92)	0.78

Note. PHQ-9: 1 = ”Not at all”, 2 = ”Several days”, 3 = ”More than half the days”; ESSI-D: 1 = ”None of the time”, 2 = ”A little of the time”, 3 = ”Some of the time”, 4 = ”Most of the time”, 5 = ”All the times”; SOC-13: for Items 1,5,6,12,13,8–10,7 double pole stages from 1 = “very seldom or never” –7 = ”very often”, Items 2,3 double pole stages from 1 = “ never happened” –7 = ”always happened again”, Item 4 double pole stages from 1 = “no clear goals or purpose at all” –7 = ”very clear goals and purpose”, Item 7 double pole stages from 1 = “a source of deep pleasure and satisfaction”–7 = ”a source of pain and boredom”, Item 11 double pole stages from 1 = “you overestimated and underestimated its importance” –7 = ”you saw things in the right proportion”; Brief COPE: 1 = “I haven’t been doing this at all”, 2 = ”A little bit”, 3 = ”A medium amount”, 4 = ”I’ve been doing this a lot”.

**Table 2 ijerph-19-16423-t002:** Bivariate and multiple logistic regression analyses concerning the independent variable mental health (absence of any PHQ-syndrome) with *n* = 1938.

Variable	Mental Health Bivariate Regression Models Unadjusted Odds Ratios (97.5% Confidence Interval; CI)	Mental Health Multiple Regression Model Adjusted Odds Ratios (97.5% CI)
**Age**	1.0937 (1.01284; 1.18055) *	**1.12095 (1.01817; 1.2341) ***
**Gender** _a_	1.54106 (1.26073; 1.88264) ***	**1.67213 (1.27904; 2.18799) *****
Social support (ESSI total score)	1.1164 (1.09042; 1.14389) ***	1.0018 (0.96756; 1.03743)
**Sense of coherence (SOC total score)**	1.12174 (1.10922; 1.13495) ***	**1.0884 (1.07384; 1.10359) *****
Coping (Brief COPE)		
Active Coping	1.22161 (1.06837; 1.39714) **	0.89599 (0.72859; 1.10092)
Planning	0.97096 (0.84962; 1.10987)	0.97874 (0.79436; 1.20587)
**Positive Reframing**	1.65703 (1.46641; 1.87514) ***	**1.3228 (1.10742; 1.58206) ****
Acceptance	1.33165 (1.17269; 1.5134) ***	1.13747 (0.95699; 1.35257)
Humor	0.94585 (0.84696; 1.0555)	0.86441 (0.74309; 1.00445)
Religion	1.03841 (0.92752; 1.16048)	1.05369 (0.91621; 1.21032)
Use of Emotional Support	1.25096 (1.11778; 1.40099) ***	1.23085 (0.98234; 1.54377)
Use of Instrumental Support	1.09925 (0.98837; 1.2226)	0.9281 (0.75648; 1.13811)
Self-Distraction	0.68161 (0.59463; 0.7801) ***	0.91954 (0.76393; 1.10616)
Denial	0.38179 (0.31175; 0.46285) ***	0.8857 (0.69954; 1.11468)
Venting	0.88574 (0.78041; 1.0042)	0.87017 (0.71686; 1.0555)
**Substance Use**	0.38179 (0.31175; 0.46285) ***	**0.38155 (0.2693; 0.52637) *****
**Behavioral Disengagement**	0.30713 (0.24831; 0.37674) ***	**0.61002 (0.47241; 0.78296) *****
**Self-Blame**	0.41021 (0.36211; 0.46318) ***	**0.73447 (0.62658; 0.85985) *****
Nagelkerke’s R²		0.4254

Note. _a_ Coding: 2 = female, 3 = male. * *p* < 0.05; ** *p* < 0.01; *** *p* < 0.001. Significant results are highlighted in bold type.

**Table 3 ijerph-19-16423-t003:** Bivariate and multiple logistic regression analyses concerning the independent variable absence major depression (absence of PHQ-depression) with *n* = 1938.

Variable	Absence of Major Depression Bivariate Regression Models Unadjusted Odds Ratios (97.5% Confidence Interval; CI)	Absence of Major Depression Multiple Regression Model Adjusted Odds Ratios (97.5% CI)
Age	1.07904 (1.00186; 1.16295) *	1.07831 (0.98085; 1.18619)
**Gender** _a_	1.31895 (1.08456; 1.60645) **	**1.40641 (1.07808; 1.8383) ***
**Social support (ESSI total score)**	1.14303 (1.11959; 1.16749) ***	**1.04865 (1.01547; 1.08318) ****
**Sense of coherence (SOC total score)**	1.14041 (1.12642; 1.15526) ***	**1.101 (1.08552; 1.11724) *****
Coping (Brief COPE)		
Active Coping	1.34126 (1.17807; 1.52908) ***	0.94995 (0.77565; 1.1634)
Planning	1.04692 (0.92212; 1.18862)	1.16245 (0.94883; 1.42502)
**Positive Reframing**	1.6883 (1.49778; 1.9068) ***	**1.3301 (1.10893; 1.5977) ****
Acceptance	1.29496 (1.14704; 1.46337) ***	1.03957 (0.87238; 1.23906)
**Humor**	0.92602 (0.83465; 1.02742)	**0.80294 (0.69323; 0.92921) ****
Religion	0.95787 (0.86114; 1.06622)	0.8799 (0.76522; 1.01206)
**Use emotional Support**	1.45612 (1.30618; 1.62513) ***	**1.28947 (1.03305; 1.61115) ***
**Use Instrumental Support**	1.18484 (1.07015; 1.31268) **	**0.81261 (0.66824; 0.98699) ***
Self-Distraction	0.73318 (0.64537; 0.83214) ***	0.97831 (0.81877; 1.16902)
Denial	0.42825 (0.36779; 0.49665) ***	0.90203 (0.74405; 1.09243)
Venting	0.99244 (0.88165; 1.11746)	0.89181 (0.74139; 1.07201)
**Substance Use**	0.46335 (0.39131; 0.54504) ***	**0.71543 (0.58113; 0.87582) ****
**Behavioral Disengagement**	0.32159 (0.27066; 0.38023) ***	**0.62581 (0.50293; 0.77651) *****
**Self-Blame**	0.38287 (0.34025; 0.42959) ***	**0.65252 (0.56171; 0.75711) *****
Nagelkerke’s R²		0.47272

Note. _a_ Coding: 2 = female, 3 = male. * *p* < 0.05; ** *p* < 0.01; *** *p* < 0. 001. Significant results are highlighted in bold type.

## Data Availability

The data that support the findings of this study are available on request from the corresponding author. The data are not publicly available due to privacy or ethical restrictions.

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
