# Peer review of "Predictors of Students’ Mental Health during the COVID-19 Pandemic: The Impact of Coping Strategies, Sense of Coherence, and Social Support"

_ijerph, 2022, doi:10.3390/ijerph192416423_

Round 1

Reviewer 1 Report

In general, I found the paper well written and well performed.

A few remarks:

“Significant findings in the bivariate analyses but not in the multiple logistic regression analyses (e.g. active coping, acceptance) might be explained by stronger third variables in the model which correlate with the dependent variable. There is a correlation of the weaker variables with the outcome but there are predictors with a better fit so that these weaker variables are consequently less predictive than others.”

Did the authors take into account multicollinaerity? Or do you feel it does not apply here? Because it might be that results differ because of a too strong correlation between predictors? Because if I’m not mistaken, do you not use the BRIEF cope predictors individually in the models? Thus, wouldn’t they correlate strongly with each other?

Another point is: isn’t there a fear of medicalization? The paper concludes with “Our results give hints that the necessity of psychotherapeutic treatment or student counselling services might be needed more frequently compared to pre-pandemic times.” But, can you reasonably state that with your data? Which is cross-sectional, online, and does not take into account other factors? Maybe there is trauma in these groups? Maybe it was due to the period you assessed the mental health? For example, was it an exam period?

I think the discussion needs to take into account such factors. I am not saying that COVID does not have a mental health impact, I am saying that clearly showing it scientifically is more difficult than just stating it does. Because there are 1000s of papers on mental health and covid, and they all assume covid has an impact on population X or Y, despite that they use cross-sectional designs, and then they refer to other papers to prove their point, despite that most of those papers are also cross-sectional designs... 

Reviewer 2 Report

Predictors of Students´ Mental Health during the Covid-19 Pandemic: The Impact of Coping Strategies, Sense of Coherence, and Social Support

The current study focused on understanding the protective role of certain factors for students’ mental health during COVID-19. It can potentially promote our understanding of protective factors for depression during the pandemic. However, there are some serious concerns that I have after carefully reading the manuscript. The most important thing is that the authors failed to provide a rational explanation as to why these protective factors are particularly important, hence chosen.

Secondly, along with the weak theoretical structure, the discussion part is very weak that I could barely grasp the value of such argument. Below are some specific comments.

In regards to the different student groups class participation, it is confusing to read. Why did certain groups have more participation than others? The authors need to clarify.

What is PHQ?

The authors need to define key concepts before specifically talking about the relationships.

There are so many protective factors, I’m curious to know why these factors are chosen by the authors to be protective factors for one’s mental health?

It is a cross-sectional study, and language such as “predict” should be refrained from.

Why is lower humor use associated with absence of depression? The authors’ seeming explanation regarding scale properties is very weak and not scientifically sound. The authors should not only provide statistical reasons for the lack of significance, or otherwise.

The explanation regarding other protective factors in relation to depression is also very weak. I have a hard time grasping the value of the current study.  

Others: Great English editing is needed for the current manuscript, for there are various places where grammatical mistakes were observed.

Author Response

Review 2:

Point#1: “In regards to the different student groups class participation, it is confusing to read. Why did certain groups have more participation than others? The authors need to clarify.”

Answer#1: We hope to clear the procedure of including cases fort he analyses with the following sentence which was added in chapter 3.1. Descriptive Statistics (l. 306-307):

„In order to include the maximum of cases in any analyses different N are reported.“

Point#2: “What is PHQ?”

Answer#2: We hope to clear this point as follows. When it is mentioned for the first time, „Patient Health Questionnaire“ is written out (line). Additonally, further information is given concerning PHQ: it was designed for „non-psychiatric patients“ (line 197) and „was developed for general medical practioners who are not psychiatrists“ (lines 200-201)

Point#3: “The authors need to define key concepts before specifically talking about the relationships.”

Answer#3: To address this issue, we added more detailed information about the key concepts (social support, sense of coherence and the salutogenesis model of Antonovsky, and coping model of Carver, Scheier, and Weintraub) in the introduction.   

„Social support can be subdivided into instrumental, informational, and emotional support. Instrumental support includes concrete actions like helping with housekeeping or shopping. Informational support means to provide important information (e.g., announcements) about opportunities to act, while emotional support can include, e.g., listening and provide encouragement [16].“ (l. 105-109)

„The salutogenetic model focuses on how to conserve actively a healthy state.“ (l. 116-117)

„In this study, the coping model of Carver, Scheier, and Weintraub was assessed. In contrast to various dichotomous coping models, Carver et al. (1989) postulated a multidimensional approach [23]. It combines the self-regulation model of Carver and Scheier as well as the transactional model of stress and coping of Lazarus and Folkman (1984) [24]. They define coping as behavioral and cognitive efforts to manage stressful events or circumstances. Carver et al. (1989) propose to not categorize most of the coping strategies as strictly helpful or not [23].“ (l. 130-136)

Point#4: “There are so many protective factors, I’m curious to know why these factors are chosen by the authors to be protective factors for one’s mental health?”

Answer#4: This study is a secondary analysis (this point is added at the beginning of 2.3. Statistical Analyses, l. 278). Unfortunately, a theoretical based selection of protective factors was not possible.

Point#5: “It is a cross-sectional study, and language such as “predict” should be refrained from.”

Answer#5: Language was adjusted (please see track changes in the manuscript).

Point#6: “Why is lower humor use associated with absence of depression? The authors’ seeming explanation regarding scale properties is very weak and not scientifically sound. The authors should not only provide statistical reasons for the lack of significance, or otherwise.”

Answer#6: To address this issue, we added the following to our discussion regarding non-significant findings and humor:

On the other hand, we did not request religiousness of the students. Consequently, we are not able to estimate if religion plays a significant role in the life of the student sample. According to the results of Carver et al. (1989), when they developed the original COPE-Questionnaire, different findings about the frequency of using single coping strategies were reported. If situational coping was examined higher engagement in planning was reported if the person estimated the event or situation as modifiable instead of something which has to be accepted [22]. We can argue that the pandemic circumstances at university were not changeable and conditional upon policy. This might have led to a reduced usage of the strategy planning. If a strategy is less used it is not able to prevent from mental strain. Venting as a situational coping strategy seems to be often used if the event matters to the person, like Carver et al. stated [22]. They argue that venting might be a helpful strategy on one hand e.g. to move further after loss of a close person, but on the other hand can hinder adjustment to distress if distress is emphasized and maintained, e.g., in the sense of rumination [22]. It could be seen as a maintaining factor. For example, Marr et al. 2022 found venting to be a mediator variable concerning chronic general anxiety disorder and 18 years later major depressive disorder and vice versa [38]. But should be taken into account that relevant dispositional coping strategies are not congruent with relevant situational coping strategies which also might have an impact on mental health or absence of major depression.  (l. 387-405)

“This might h been the case for the situational coping strategy humor and make it difficult to interpret. Lower humor was associated with the absence of major depression exclusively in the multiple logistic regression analysis, so it is not interpretable. In a swiss study, results showed that the effectiveness of coping with stressful events and resulting negative emotions depends on the type of humor. If humor is positive, it has a positive impact on down- regulating negative emotions and foster positive emotions. If humor is mean and negative, the converse effect was found [39]. In the Brief COPE, the items do not specify the type of making jokes or fun about the situation. It is possible that the students used the more detrimental style of humor.” (l. 408-417)

Point#7: “The explanation regarding other protective factors in relation to depression is also very weak. I have a hard time grasping the value of the current study. “

Answer#7: To address this issue, we added the following in our discussion:

„Kugbey et al. (2015) found social support of friends and significant others to relevant to the level of depression in college students while family support seems to play an important role for perceived stress but not for level of depression. We did not examine the source of social support. There might occur different effects on mental health in general or other mental disorders [37]. Especially, contact to peers and friends was highly restricted because it was allowed to meet more people if they lived in the same household in Germany which was mostly not the case with friends.“ (l. 368-374)

Point#8: Others: Great English editing is needed for the current manuscript, for there are various places where grammatical mistakes were observed.

Answer#8: We are greatful for this hint. We have checked English language and style by a native speaker.

Reviewer 3 Report

First of all, congratulate the authors for the work done, however, there are certain aspects that need to be improved. So, I list them below:

1. Since this is a quantitative study, instead of research questions, I suggest that you indicate the hypotheses of your study.

2. The method should include the Procedure section (add how the questionnaires were administered and how the confidential treatment of the data was guaranteed, as well as whether the research has the favorable report of any Committee).

3. The coefficient to measure the reliability of the scales used (Cronbach's alpha) is not shown, which is why I ask you to include it.

4. The discussion is a comparison between their results and those found by other authors cited previously in the introduction.

Reviewer 4 Report

- I apologize for my level of English. It's not very good.

- I congratulate the authors of this study for the relevance of the selected topic. Mental health and its relationship with the Covid pandemic is highly topical and of great interest.

- The number of participants in the study is high and that is a strength of the study.

However, I have doubts about the actual number of participants, given by a part in line 268 of 3.1. Descriptive Statistics, indicates that the sample is composed of (N= 2,398) consisted of 780 male (32.53%) and 1,528 female (65.8%).

However, in section 2.1 Procedure, on line 164-165 indicates: The final sample for analyzes included N=1,938 male and female students.

Also, in Table 1, Major depression (PHQ-9), n= 2,139.

I think the N is not entirely clear.

Round 2

Reviewer 2 Report

I saw that the authors have made effort to work on many pieces of feedback. The discussion part has been greatly extended in terms of explaining their findings. Unfortunately, along with my first major point, the introduction part is still not persuading enough for me as the opening of this manuscript. 
